# Effects of Degradation on Microbial Communities of an Amazonian Mangrove

**DOI:** 10.3390/microorganisms11061389

**Published:** 2023-05-25

**Authors:** Gleyciane Machado da Costa, Sávio Souza Costa, Rafael Azevedo Baraúna, Bruno Pureza Castilho, Izabel Cruz Pinheiro, Artur Silva, Ana Paula Schaan, Ândrea Ribeiro-dos-Santos, Diego Assis das Graças

**Affiliations:** 1Laboratory of Biological Engineering, Guamá Science and Technology Park, Belém 66075-750, Brazil; gleyci.ane2219@gmail.com (G.M.d.C.); savscosta@gmail.com (S.S.C.); rabarauna@ufpa.br (R.A.B.);; 2Laboratory of Genomics and Bioinformatics, Center of Genomics and Systems Biology, Institute of Biological Sciences, Federal University of Pará, Belém 66075-110, Brazil; 3Laboratory of Medical and Human Genetics, Institute of Biological Sciences, Federal University of Pará, Belém 66075-110, Brazil; apschaan@gmail.com (A.P.S.); akelyufpa@gmail.com (Â.R.-d.-S.)

**Keywords:** 16S rRNA, deforestation, anthropogenic impact, microbiome

## Abstract

Mangroves provide a unique ecological environment for complex microbial communities, which play important roles in biogeochemical cycles, such as those for carbon, sulfur, and nitrogen. Microbial diversity analyses of these ecosystems help us understand the changes caused by external influences. Amazonian mangroves occupy an area of 9000 km^2^, corresponding to 70% of the mangroves in Brazil, on which studies of microbial biodiversity are extremely scarce. The present study aimed to determine changes in microbial community structure along the PA-458 highway, which fragmented a mangrove zone. Mangrove samples were collected from three zones, (i) degraded, (ii) in the process of recovery, and (iii) preserved. Total DNA was extracted and submitted for 16S rDNA amplification and sequencing on an MiSeq platform. Subsequently, reads were processed for quality control and biodiversity analyses. The most abundant phyla were Proteobacteria, Firmicutes, and Bacteroidetes in all three mangrove locations, but in significantly different proportions. We observed a considerable reduction in diversity in the degraded zone. Important genera involved in sulfur, carbon, and nitrogen metabolism were absent or dramatically reduced in this zone. Our results show that human impact in the mangrove areas, caused by the construction of the PA-458 highway, has resulted in a loss of biodiversity.

## 1. Introduction

Mangroves are one of the most dynamic and productive ecosystems and have great ecological importance. Located along tropical and subtropical regions, they constitute more than half of the terrestrial coastline forming a transitional environment between sea and land [1]. Mangroves are subjected to periodic tidal flooding, which results in variable salinity, redox potential, and anaerobic/aerobic conditions [2,3]. These characteristics make mangroves capable of harboring highly diverse microbial communities, which in turn are responsible for many essential features of biogeochemical cycles [4,5], thus microbial communities play a central role in maintaining the health and equilibrium of the mangrove environment. 

The nitrogen cycle in mangroves is mediated by diverse groups of microorganisms and involves the transformation of nitrogen in various forms through key processes, such as nitrogen fixation, nitrification, denitrification, and ammonification [4]. Microbial communities in mangrove soils play a critical role in the phosphorus (P) cycle by mediating key processes involved in P cycling, such as (re)mineralization, immobilization, and adsorption, which influence P availability and cycling in mangrove ecosystems [5]. Microorganisms also mediate many sulfur transformation reactions, such as sulfide oxidation, sulfate reduction, and sulfur disproportionation [6]. The productivity of mangroves fuels ecosystem services based on plant products and makes up the foundation of the ecosystem food web.

Brazil has the second largest area of mangroves in the world [7]. It was reported that the mangroves in Brazil possess 8.5% of the world’s carbon stocks found in these ecosystems, most of which is in the upper meter layer of soil, making this biome crucial for maintaining the climate [8]. Amazonian mangroves occupy an area of more than 10,000 km^2^ and account for 80% of the country’s mangroves. Microbes play a major role in the carbon cycle, but studies focused on microbial diversity in the Amazonian region are extremely scarce, despite reports stating these areas are at risk due to the consequences of human activities [7,9,10].

The main bacterial phylum found in mangrove samples is Proteobacteria, which is considered part of the core microbiome, but other groups are also frequently found such as **Planctomycetes, Acidobacteria, Bacteroidetes, and Chloroflexi** [11]. Previous research in Brazil has aimed to understand and characterize the role of microbial diversity in mangrove areas [12,13,14,15]. In São Paulo state, evidence shows that Proteobacteria, Actinobacteria, Firmicutes, and Bacteroidetes are more abundant in areas impacted by humans and oil spills [16]. In this same location, it was also reported that mangrove zones are richer in taxonomic and functional diversity [15]. 

Nóbrega et al. [9] reported a high abundance of Proteobacteria and Actinobacteria in water samples from an Amazonia mangrove area, with functional profiles related to carbon, nitrogen, sulfur, and methane cycles. In the same region, Tavares et al. [10] showed a high abundance of Proteobacteria, and that the microbiomes associated with *Rhizophora mangle* root soils are similar to those from the Brazilian southeast, in spite of their geographical distance. 

The PA-458 highway was constructed to connect Bragança city to the coastal and touristic beach area of Ajuruteua, leading to the destruction of Amazonian mangrove areas along its path [17]. This impact affected not only aquatic systems that depend on the exchange of nutrients [18] but also animal nutrition and the riverine subsistence economy [19,20]. There are zones within this mangrove area that are considered to be under ecological and environmental recovery due to recent measures aimed at preservation, such as community environmental awareness programs, research, reforestation with native plants, and others. 

Studies assessing the environmental impacts on the microbiome of mangroves are important to aid in the preservation and restoration of these valuable ecosystems. In this study, we characterize and compare changes in the microbial community structures in mangrove sediments impacted by the construction of the PA-458 highway, in the Brazilian Amazon, by analyzing preserved and under-recovery areas through amplicon sequencing.

## 2. Materials and Methods

### 2.1. Sample Collection and Site Description

The Bragança Region is located on the Amazonian Atlantic coast, and its mangrove peninsula encompasses the estuaries of the Caeté and Tapera-Açu rivers and is considered the second largest continuous mangrove ecosystem in the world. Soil compaction for the construction of the highway interrupted the flow of water and nutrients through the mangrove in several areas, in addition to streams, thus impacting the landscape, vegetation, and fauna. The area experiences a semidiurnal macrotidal pattern, with tides ranging from four to six meters [19]. The water salinity varies from zero to as high as 100 parts per thousand (ppt) throughout the annual cycle. Mangrove samples were collected in triplicate at six points along the PA-458 highway (Figure 1). The collection points were separated on the East and West sides of the highway, making a total of 18 samples obtained in August 2018, during the dry season. No rain occurred during sampling.

All three collection points on the east side of the highway (P2, P4, and P6), and P5 were considered preserved, due to constant water flow and wood density where *Rhizophora mangle* L., *Avicennia germinans* (L.) L., and *Laguncularia racemosa* (L.) C. F. Gaertn predominated. On the west side, sample P1 was classified as degraded, due to the absence of vegetation and no water flow. Point P3 was considered as under-recovery because the area is under the influence of a stream and vegetation is undergrowth. 

At each point (from P1 to P6), three replicates of 30 g of subsurface sediment (~20 cm depth) were collected using a sterile soil core sampler. Vegetation was removed and soil was immediately transferred to a 50 mL tube. Samples were chilled on ice until arrival at the laboratory and then stored at −80 °C until further procedures. Due to DNA degradation during transportation, five replicates were removed for further analysis. 

Another three sediment sample replicates were collected from each point (P1to P6), to give a total of 18 samples. Replicates from each point were prepared for physicochemical analysis, which was carried out at the Brazilian Agricultural Research Corporation (EMBRAPA). For the determination of: P, K, and Na content, extraction was performed with Mehlich 1 solution (12.5 mM H_2_SO_4_ + 50 mM HCl); Ca, Mg, and Al, extraction with 1 M KCL solution; H and AI (potential acidity), extraction with calcium acetate pH 7 solution; organic matter (OM), potassium dichromate; and total nitrogen, sulfuric/Kjeldahl digestion.

### 2.2. DNA Extraction and Sequencing of 16S rRNA Genes

Total DNA was extracted from 500 mg mangrove samples using the Powersoil DNA Isolation Kit (MO BIO, Carlsbad, CA, USA) according to the manufacturer’s protocol. Eluted DNA was quantified with fluorometry on a Qubit 4.0 (Thermo Scientific, Waltham, MA, USA) and subsequently stored at −20 °C. Library preparation was carried out according to the Illumina Metagenomic Sequencing Library Prep protocol with established primers and Illumina Nextera adapters targeting the V3–V4 region of the 16S rDNA, as follows: Bakt_341F (CCTACGGGNGGCWGCAG) e Bakt_805R (GACTACHVGGGTATCTAATCC), using the following cycle: 95 °C for 4 min, followed by 25 cycles at 94 °C for 1 min, 60 °C for 1 min, and 72 °C for 2 min. The PCR reaction final volume was 25 µL, containing 2.5 μL of DNA (5 ng/μL), 5 μL of each primer (1 μM), and 12.5 μL of 2x KAPA HiFi HotStart ReadyMix. Purification and size selection of PCR products were performed using a Pronex^®^ Size-Selective Purification System (Promega, Madison, WI, USA). Libraries were subsequently quantified with TapeStation (Agilent, Santa Clara, CA, USA) using High Sensitivity D1000 screen prior to sequencing on the Illumina MiSeq platform, using Reagent Kit v2 (500 cycles, 2 × 250 bp) on an Illumina MiSeq sequencer (Illumina, San Diego, CA, USA). 

### 2.3. Data Analysis

Read quality was evaluated using FASTQC v0.12.1 [21], low-quality reads smaller than 200 bp were filtered out, and chimeras were removed using USEARCH [22]. After quality control analyses, one replicate yielded less than 5000 raw reads with low-quality sequences and was removed from further analysis. Reads were grouped into Operational Taxonomic Units (OTU) with 97% similarity and taxonomy was assigned to each OTU (excluding singletons) by performing BLAST searches in the SILVA v.132 database [23]. Alpha and beta diversity analyses were performed in R v.3.5.343 using Phyloseq [24], Vegan [25], and LEfSe [26] packages.

Statistical differences between alpha diversity results were tested using T-Test. To perform beta diversity analyses, sample counts were normalized based on their relative abundances, and PERMANOVA was used to examine the Bray–Curtis Distances. Heatmap and sample clustering analysis were performed using the Wilcoxon Rank Sum test, and the taxonomic differences between microbial communities were obtained using LDA Effect Size (LEfSe) with a *p*-value 0.1 using the Kruskal Wallis test.

## 3. Results

The data presented in Table 1 correspond to the results of the physicochemical analyses of mangrove sediment samples. Samples were acid to neutral (pH range 5.1–6.4). We observed a difference in the amount of carbon, OM, phosphorus, potassium, and cation-exchange capacity (CEC) when comparing degraded mangrove sediment to the preserved areas (except for P6), but without in-depth statistical analysis.

We obtained 999,997 high-quality reads (per sample minimum = 43,328, maximum = 131,784, mean = 83,333, median = 84,411, standard deviation: 25,795). We observed 14,155 different OTUs throughout all samples and considered that sequencing efforts were sufficient due to rarefaction curves reaching a plateau of around 25,000–30,000 sequences in library size (Appendix A). It is important to note that the number of samples taken from each zone was unbalanced, which can affect the strength of the inference.

Alpha diversity results (Table 2) revealed that samples from degraded mangroves showed a dramatically lower species richness (~2.5× less OTUs) when compared to others, with a mean Shannon index of 5.55 (mean of 1759 OTUs), while preserved samples showed a mean Shannon index of 7.00 (mean of 4496 OTUs), and samples from the recovery area reached a Shannon index of 7.12 (mean of 4653 OTUs). The Simpson index shows a high degree of diversity and evenness in all samples. 

Alpha diversity results revealed significant differences (*p* < 0.05) in microbial diversity between the environment impacted by the road construction (degraded) and both the preserved and the recovery samples, demonstrated by lower Chao1, Shannon, and Simpson diversity indexes (Figure 2). 

The most abundant phyla were Proteobacteria, Firmicutes, and Bacteroidetes in all three mangrove sites, however, in notably different proportions (Figure 3A). Alphaproteobacteria were assigned mostly to orders Rhizobiales, Rhodospirillales, and Rhodobacterales. Deltaproteobacteria were assigned mostly to orders Desulfobacterales, Syntrophobacterales, and Myxococcales, while Gammaproteobacteria were classified mostly in Legionellales, Chromatiales, and Methylococcales. Alpha, Delta, and Gammaproteobacteria represented more than 90% of the reads assigned in Proteobacteria phylum, but Beta, Epsilon, and Zetaproteobacteria were also observed, as well as Oligoflexia class.

Proteobacteria represented 38–55% of relative abundance in preserved and recovery areas, but in the degraded mangrove was reduced to 12–15%. On the other hand, Firmicutes phylum was observed in significantly higher abundance (30–35%) in the degraded area when compared to other points, as well as Bacteroidetes (25–29%), mainly represented by Prevotellaceae, Lachnospiraceae, and Ruminococcaceae families. Furthermore, Choloroflexi, Acidobacteria, Actinobacteria, Verrucomicrobia, and Planctomycetes were found in all areas (less than 10% of relative abundance), and were more abundant in preserved and under-recovery areas than in the degraded zone (Figure 3A). 

Considering the differentially abundant taxa (LDA analysis), Firmicutes, Bacteroidetes, Elusimicrobia, Lenthisphaerae, Tenericutes, and Acetothermia were significantly more abundant in the degraded zone (Figure 3B), while Acidobacteria, Actinobacteria, Planctomycetes, Latescibacteria, and Nitrospirae were higher in the recovery area. The preserved zone presented a significantly higher abundance of Proteobacteria, Chloroflexi, Ignavibacteriae, and Deferribacteres.

Beta-diversity analyses (PCoA and dissimilarity) showed that samples collected in the degraded zone form a distinct group when compared to the preserved and in-recovery areas (Figure 4B), with axis 1 explaining 78.6% of the variability. In the Heatmap plot (Figure 4A), we observed that Thermodesulfobacteria, Acetothermia, Tenericutes, Bacteroidetes, Lentisphaerae, Elusimicrobia, and Firmicutes are more abundant in the degraded zone, while the other taxa are in higher proportion in preserved and recovery areas. 

## 4. Discussion

Mangrove ecosystems face global threats from multiple factors, such as coastal development, pollution, and increasing sea levels. This study shows a dramatic loss of microbial phylogenetic diversity in zones affected by the construction of a 40 km long highway when compared to preserved or areas under recovery. The preserved and recovery zones have a higher concentration of carbon and OM, receiving nutrients from the riverine entrance and tide, as well as having a higher microbial diversity. The abundant vegetation and fauna contribute to a richer environment, with more niches and microbes being important players in this scenario.

Among the species most found in Brazilian mangroves, 53–60% belong to the phylum Proteobacteria [27], which corresponds to about half of the bacterial genomes recorded at the National Center for Biotechnology Information (NCBI), corroborating the results of this study [28,29,30]. Represented mostly by the classes of Deltaproteobacteria and Gammaproteobacteria, Proteobacteria are part of the core microbiome of mangroves worldwide [31,32,33]. Proteobacteria are found in freshwater and marine environments in all latitudes, where they are responsible for many roles, including the degradation of OM [9,16].

In a study carried out on three mangrove areas in the state of São Paulo (contaminated by oil, under anthropogenic effect, and preserved), more than 45% of all microbial diversity was represented by Proteobacteria in all three areas [16]. The same study showed a percentage below 15% for Firmicutes, and even lower abundances for Bacteroidetes in mangrove areas under the effect of anthropogenic action, despite the former being reported as the second most abundant phylum after Proteobacteria, followed by Actinobacteria and Bacteroidetes. In our observations, the Firmicutes phylum was detected in all mangrove environments, but in higher abundances in the degraded zone. A higher abundance of the phyla Firmicutes and Bacteroidetes was also found in mangroves contaminated by sewage on the island of Hainan in China [34]. In contaminated mangroves along the Red Sea Coast in Saudi Arabia, Bacteroidetes showed a relative abundance of >1% compared to the pristine zones (<1%) [35].

In our study, Firmicutes and Bacteroidetes are significantly more abundant in impacted mangroves when compared to pristine zones (preserved and recovery), and are mainly represented by Prevotellaceae, Lachnospiraceae, and Ruminococcaceae families, whose members are typically found in gut and rumen microbiomes. One explanation for this could be the intense populational flow and human activity throughout the highway extension. It is worth mentioning that the mangrove areas impacted in the studies discussed here had a small amount of vegetation in their environments. A study of the vertical distributions of rhizosphere bacteria from three mangrove species in the Beilun Estuary located in southern China revealed that plant species and depth played a role in the formation of microbial communities [29].

The phylum Acidobacteria represents a group of soil bacteria, whose members are widely and evenly distributed throughout almost all ecosystems [36], mainly in acidic soils. These bacteria may have an important contribution to the mangrove ecosystem, as they are phylogenetically diverse and may play many roles in various biogeochemical cycles, such as those of nitrogen, carbon, and sulfur, due to their metabolic versatility [37]. In the present study, many Acidobacteria subgroups (1, 2, 3, 6, 16, 22, and 23) were found in higher abundance in the under-recovery zone, but in lower proportion in the highly impacted mangrove, which is expected for an area with biodiversity loss [38,39]. These subgroups were found in Brazilian mangrove sediments associated with *Rhizophora mangle* [10] and in Restinga soils from the Brazilian southeast region [15]. 

Verrucomicrobia phylum represented less than 2% of the reads in all zones, but in degraded samples was nearly absent. This phylum is frequently found in low frequency in Brazilian mangroves [10,16,27,40], as well as worldwide [29]. Their role in mangrove sediments is poorly understood, but in the soil, they regulate methane emissions [41] and participate in nitrogen transformations [42], which reinforces the hypothesis that the rare microbiome plays a crucial role in mangrove cycles. 

The Archaea domain was not considered abundant (<2%) as reported in other studies [43], but we observed that the degraded zone had a lower abundance of archaea than preserved areas. Euryarchaeota were represented mainly by Halobacteria and Methanogenic archaea, while Crenarchaeota were all represented by the Thermoproteales and Desulfurococcales. It is known that Crenarchaeota is also the major player in the nitrogen cycle in acidic soils (pH < 5.5) [44], which is the pH range of samples in this study. These groups are important in the carbon and sulfur cycle, which is probably affected in the degraded zone. We also observed ammonia-oxidizing archaea (AOA) at preserved and recovery zones, assigned to the Thaumarchaeota phylum, but in low frequency (<0.1%).

Ignavibacteriae were found in low abundance (less than 3%) in preserved and recovery zones, while nearly absent in degraded samples. It is a relatively recently described phylum, which is known to be Gram-negative, anaerobic, or microaerophilic [45]. They are generally considered to be slow-growing and are often found in environments such as freshwater sediments, subsurface environments, and gastrointestinal tracts of animals. Ignavibacteria were found to be present in low abundances (<1.5%) in mangrove samples from Panama, however, were observed in higher proportions in samples from more preserved and unpolluted areas [46]. These findings are similar to our results, and suggest Ignavibacteria are probably involved in some process only possible in a preserved environment.

The phylum Latescibacteria, previously known as WS3, can be found in various environments such as terrestrial, aquatic, marine, and mangrove habitats [47,48]. They are typically Gram-negative, anaerobic, and often found in low abundance, but their ecological roles and physiological characteristics are not yet fully understood. In mangrove sediments of Goa (India), Latescibacteria were found in higher proportions in environmentally rich niche samples [28].

At the family level, we observed differential distribution patterns in the degraded and preserved or in recovery mangrove regions, including the loss of facultative or strictly anaerobes Anaerolineaceae, Desulfobacteraceae, Desulfobubaceae, Sytrophaceae, and Planctomycetaceae in the impacted area, which is frequently found in aquatic (marine or freshwater) environments. Desulfobacteraceae, Desulfobubaceae, and Sytrophaceae are strictly anaerobic bacteria that reduce sulfate to obtain energy, and some members are also able to reduce nitrate. Recently, some researchers proposed to cluster these families into a phylum named Desulfobaterota [49,50]. The Anaerolineaceae family are anaerobic bacteria found in the upper zones of mangrove sediments that play a role in nitrate reduction to ammonium [51]. In preserved and recovery zones, we observed a high abundance of Thioalkalispiraceae, which are chemolithoautotrophic bacteria able to oxidize sulfur compounds and that have been obtained from soda lakes, hypersaline lakes, or marine environments, with specific NaCl concentration requirements for growth [52]. Bacteria assigned to Planctomycetaceae are facultative aerobes typically found in aquatic and wetland zones involved in the nitrogen and carbon cycle [16,53]. Here, we consider that the absence of these bacterial groups is due to mangrove degradation, affecting nitrogen, sulfur, and carbon cycles.

Actions to recover the Bragança mangrove have been carried out in recent decades and apparently had positive results on the resilience of the microbial community in the mangrove sediment. The beta diversity analysis showed that the recovering mangrove areas share similar taxonomic diversity to preserved collection sites, with no significant difference between the groups. In addition, we also observed several differentially abundant phyla, but no statistical significance was revealed in our study. 

We observed 47 different phyla across all samples, only six of which displayed abundances greater than 5%. More than 30 phyla were considered rare or of low abundance, indicating the environment is capable of harboring a wide range of diversity that is potentially crucial for ecological balance [54]. Wang et al. (2022) [54] reported that in mangrove samples, the influence of abiotic variables on the structure of rare taxa is limited, explaining only a small portion of the observed variation. This suggests that there may be other unknown and more complex factors shaping the composition of rare taxa, and more studies focused on these groups are needed. 

The significance of this study is to show the extent of the impact caused by road construction (and potentially other anthropogenic activities) on mangrove ecosystems, which also influence microbial diversity and possibly the function of mangrove ecosystems. Our findings indicate that degraded mangrove sediments have significantly lower bacterial diversity, with a decrease in taxa responsible for sulfur, nitrogen, and carbon cycles compared to preserved and recovering mangroves. Furthermore, we showed an increase of typical human gut microbial taxa in the degraded zone, which may indicate human contamination in mangrove sediments. Studies using a whole metagenomic sequencing approach with dense and longitudinal sampling may allow for further insights into the long term impact of human activity on mangroves as well as the pace at which these ecosystems are able to recover and restore microbial communities. 

## Figures and Tables

**Figure 1 microorganisms-11-01389-f001:**
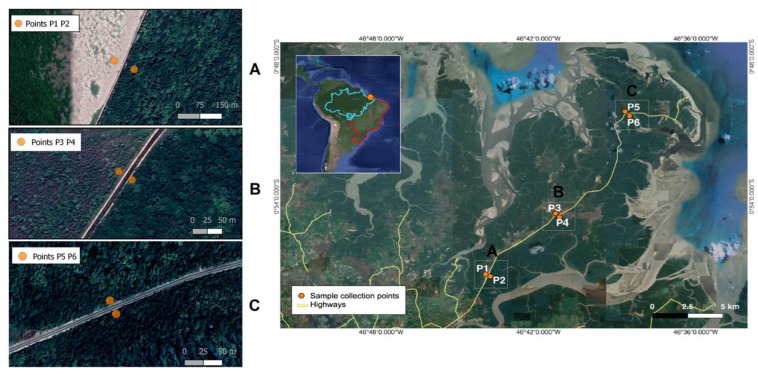
Location of each sampling point along highway PA-458 in relation to Brazil (red line) and the Brazilian Amazon territory (blue line). The (**A**) zone shows the degraded area on the left side of the road (P1), as the right side is preserved (P2). The (**B**) zone shows the area under recovery on the left side of the road (P3), as the right side is preserved (P4). The (**C**) zone shows two preserved areas on both sides of the road (P5 and P6).

**Figure 2 microorganisms-11-01389-f002:**
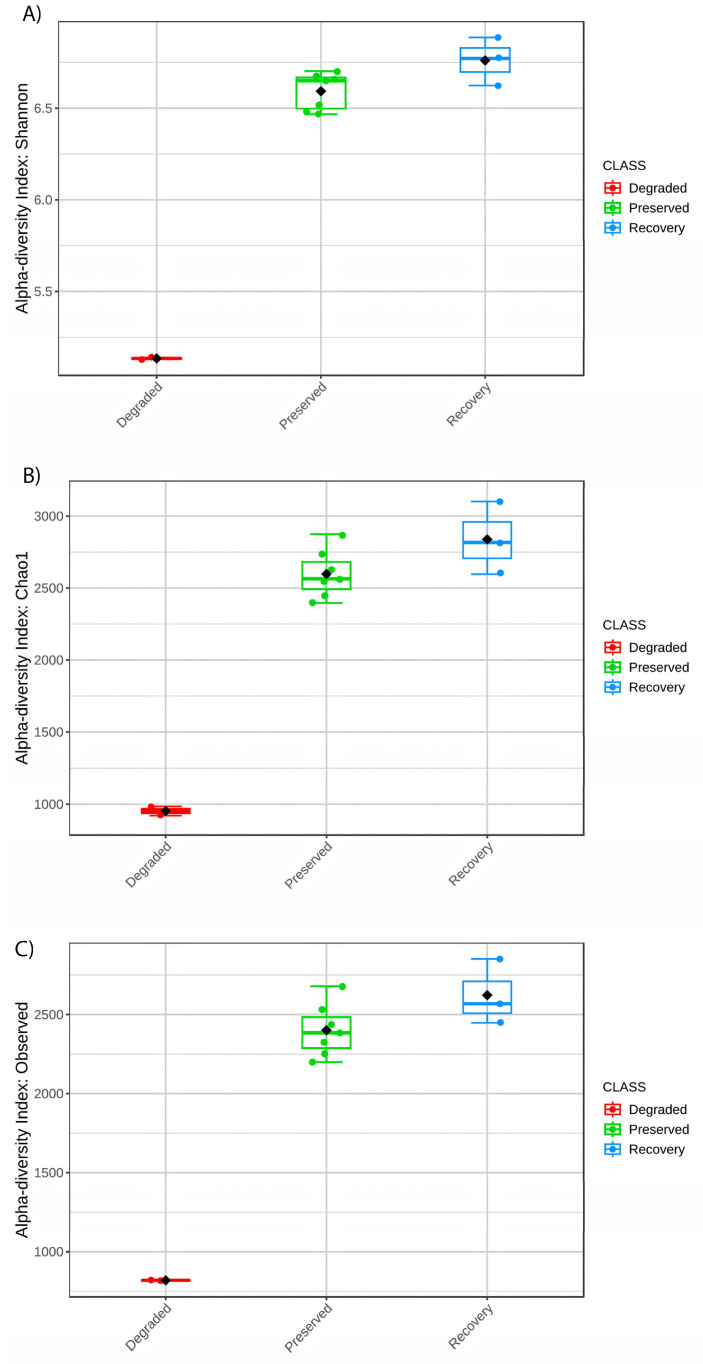
(**A**) Shannon alpha diversity index, (**B**) Chao1 alpha diversity index, and (**C**) Observed OTUs of all mangrove areas.

**Figure 3 microorganisms-11-01389-f003:**
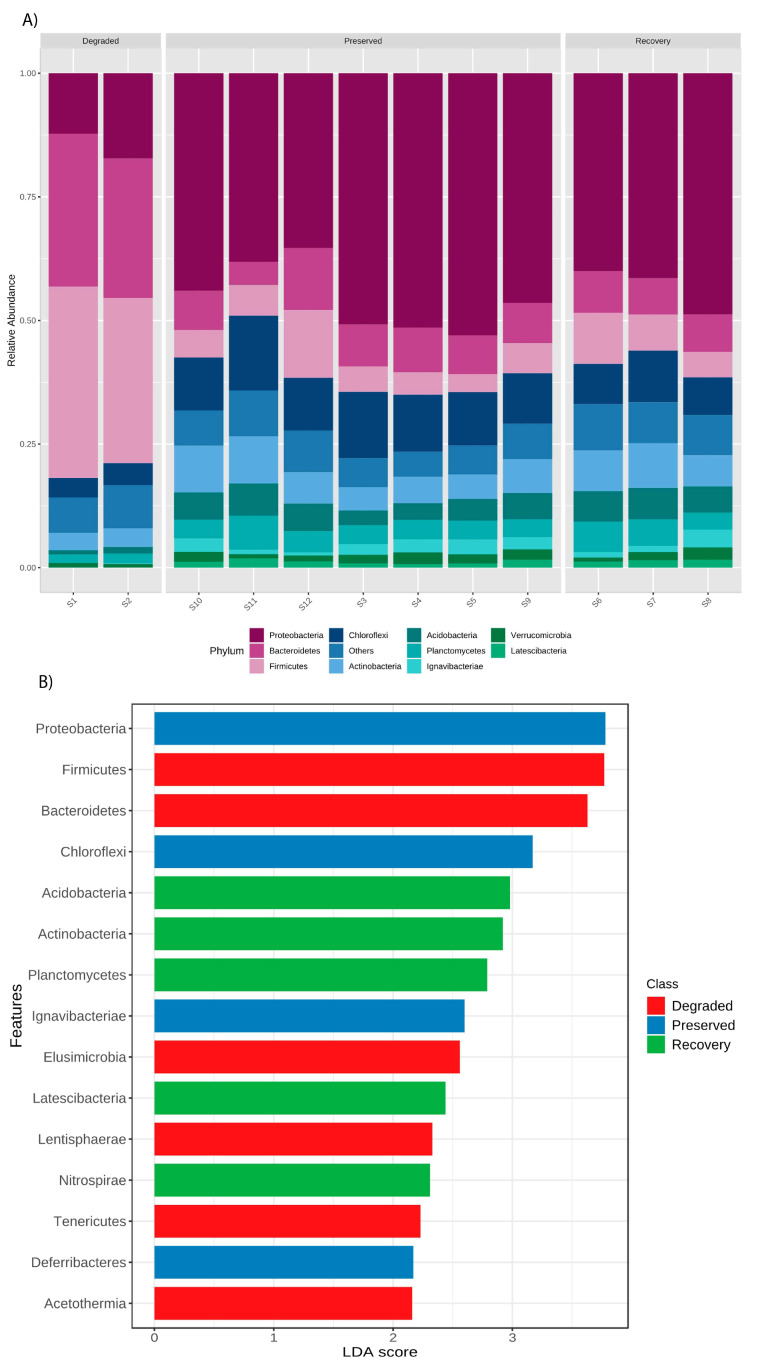
(**A**) Relative abundance at phylum level in mangrove environments. (**B**) Differentially abundant phyla according to mangrove preservation category.

**Figure 4 microorganisms-11-01389-f004:**
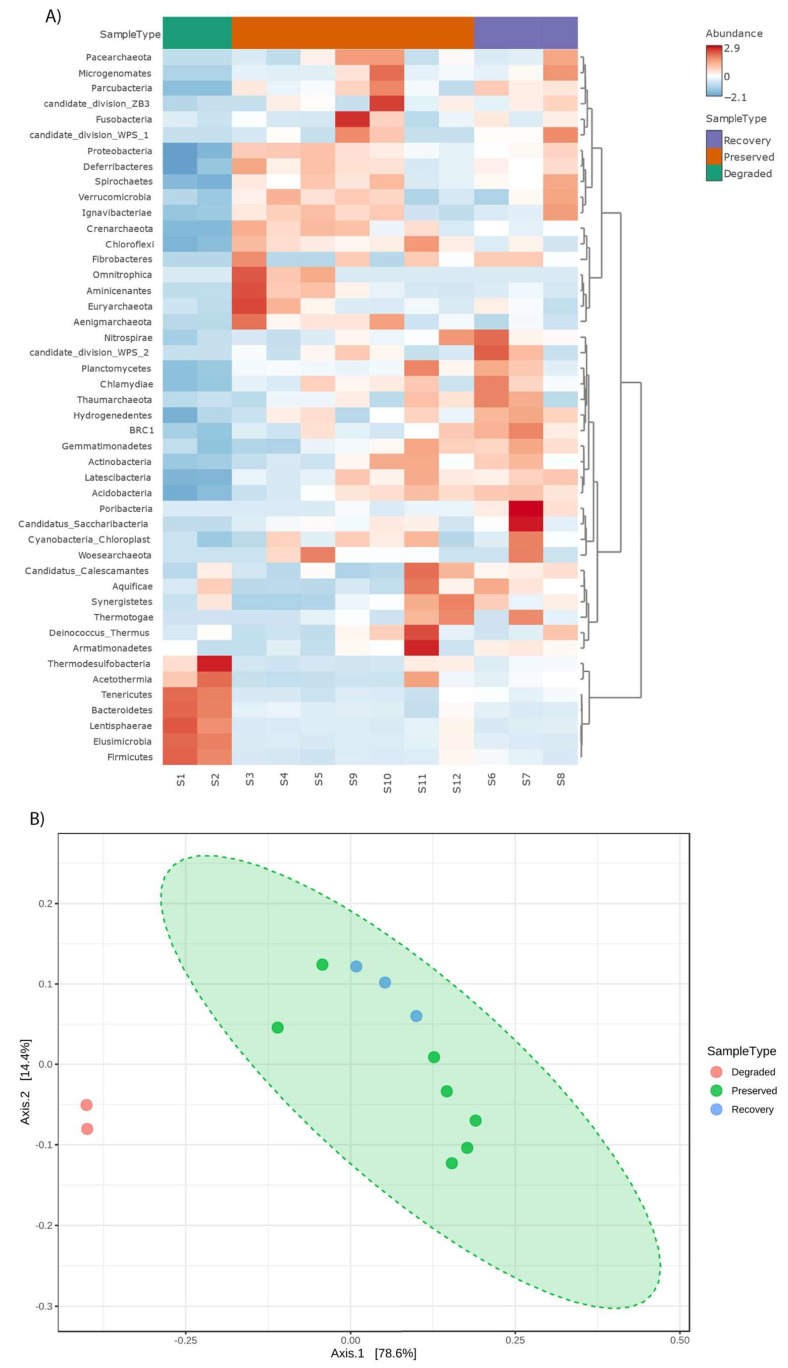
(**A**) Heatmap showing phylum abundances. (**B**) Principal Coordinate Analysis (PCoA) of Bray–Curtis distances between different collection points in the Amazonian Mangrove. Green dotted area shows the cluster of preserved and recovery samples.

**Table 1 microorganisms-11-01389-t001:** Physicochemical parameters table. Carbon (C), organic matter (OM), total nitrogen (N), phosphorus (P), potassium (K), sodium (Na), aluminum (Al), calcium (Ca), magnesium (Mg), pH and potential acidity (H + Al) and saturation. The values of each point (P1, P2, P3, P4, P5, and P6) were obtained after mixing three replicates of each.

Samples	C	OM	N	N	Ratio	P	K	Na	Al	Ca	Ca + Mg	pH	H + Al	CEC		Saturation	
ID	g/kg	g/kg	%	g/kg	C/N	mg/dm^3^			cmol/dm^3^			Water	cmolc/dm^3^	Total (cmol/dm^3^)	Effective (cmol/dm^3^)	Base (V%)	Aluminum (m%)
P1 (Degraded)	10.8	18.7	0.08	0.81	13.3	91	1303	23,072	0	5	33.2	5.9	0	136.81	136.83	100	0.01
P2 (Preserved)	16.5	28.5	0.07	0.73	22.8	16	444	3957	0.1	2.5	8.5	6.4	3.82	30.66	26.91	87.55	0.26
P3 (Recovery)	10.4	18	0.1	1.03	10.1	85	1391	19,967	0	6.1	29.9	5.6	0.49	120.75	120.29	99.6	0.02
P4 (Preserved)	24.8	42.8	0.13	1.29	19.2	43	788	10,957	0	3.6	17.4	5.8	2.09	69.14	67.07	96.97	0.04
P5 (Preserved)	21.7	37.5	0.07	0.73	30	34	919	11,976	0	3.2	15.7	5.1	0.79	70.93	70.18	98.89	0.04
P6 (Preserved)	8.4	14.5	0.07	0.7	11.9	28	876	12,082	0	3.1	15.2	5.5	0.9	70.89	70.03	98.73	0.04

**Table 2 microorganisms-11-01389-t002:** Sequencing and alpha diversity indexes results.

Sample	Collection Point	Mangrove Zone	Library Size	OTUs	Chao1	Shannon	Simpson
S1	P1	Degraded	90,508	1846	2221.686	5.642208	0.9890705
S2	P1	Degraded	131,794	1672	2213.535	5.476673	0.9883609
S3	P2	Preserved	72,723	4307	5320.197	6.945422	0.9969483
S4	P2	Preserved	93,865	4876	6218.262	7.089495	0.9973286
S5	P2	Preserved	65,004	4663	5628.641	7.091619	0.9972898
S6	P3	Recovery	43,328	4491	5379.287	7.155558	0.9974265
S7	P3	Recovery	124,523	5185	6733.596	7.296043	0.9975854
S8	P3	Recovery	89,942	4284	5545.846	6.936312	0.9966584
S9	P4	Preserved	80,006	4738	5771.278	7.065384	0.9965857
S10	P4	Preserved	84,816	4634	6056.463	7.051824	0.9972325
S11	P5	Preserved	67,836	4125	5065.642	6.937517	0.9961980
S12	P5	Preserved	55,652	4140	5362.078	6.842138	0.9962226

## Data Availability

Data are available at SRA under the IDs PRJNA934321 and PRJNA934349.

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
