# Peer review of "Effects of Degradation on Microbial Communities of an Amazonian Mangrove"

_microorganisms, 2023, doi:10.3390/microorganisms11061389_

Round 1
Reviewer 1 Report
Manuscript no.: Μicroorganisms-2329836
First author: Gleyciane Machado da Costa
Title of paper: Effects of degradation on microbial communities of an Amazonian mangrove
A. General comments
I feel that the manuscript contains interesting information’s. The visibility of figures is problematic. The manuscript has merit but it requires explanations, modifications before it is suitable for publication (see specific comments).
B. Specific comments
Title
Is OK
Abstract
I suggest authors to give some more information on the methodology.
Keywords
Are OK.
1. Introduction
The Introduction should state more clearly the reason for doing the work for the international community.
2. Materials and methods
2.3. Data Analysis
Author(s) should provide details. Please rewrite shorter and clearly the chapter.
3. Results
Author(s) should provide better the novelty places of the results with more details.
4. Discussion
Author(s) need to discuss better the novelty places of the results.
Conclusions??
Author(s) need to write clearly a sub-chapter only with the main conclusions.
References
The authors need to cheque the references in the text and in the list.
Minor or Moderate editing of English language required.
Author Response
General Response: Dear reviewer, thank you for accepting to review our manuscript.
Of the six major points evaluated by the reviewer, one was considered sufficient, and the other five were recommendations for improvement, which we tried to do as best as we can. We hope that the final version attends the recommendations.
Point 1: I feel that the manuscript contains interesting informations. The visibility of figures is problematic. The manuscript has merit but it requires explanations, modifications before it is suitable for publication (see specific comments).
Response 1: We believe that the resolution of the figures may have been compromised when the system generated the pdf file. Our figures were generated using high standard softwares (Photoshop and R packages such as phyloseq and ggplot). The files have more than 300 Mb in TIFF format, and we believe that the final version will be better.
Point 2: Abstract: I suggest authors to give some more information on the methodology.
Response 2: The abstract is limited to 200 words and we could not detail much more, but we did some modifications.
Point 3: Introduction: The Introduction should state more clearly the reason for doing the work for the international community.
Response 3: We believe that the last paragraph makes clear the objective of the study.
Point 4: Data Analysis: Author(s) should provide details. Please rewrite shorter and clearly the chapter.
Response 4: We provided as much detail as possible to allow the reproducibility of the study. The journal is rigorous in the instructions about this.
Point 5: Results: Author(s) should provide better the novelty places of the results with more details.
Response 5: We agreed. Some modifications were done to this topic.
Point 6: Discussion: Author(s) need to discuss better the novelty places of the results.
Response 6: We agreed. Some modifications were done to this topic.
Point 7: Conclusions?? Author(s) need to write clearly a sub-chapter only with the main conclusions.
Response 7: We agreed. Some modifications were done to this topic to make it clearer.
Point 8: References: The authors need to check the references in the text and in the list.
Response 8: Thank you. We checked again and some were absent.
Point 9: Minor or Moderate editing of English language required.
Response 9: Thank you. We revised the manuscript again and many improvements were done.
Points considered ok by the reviewer: Title, Keywords.
Response 10: Thank you.
Reviewer 2 Report
The article has a relevant topic. It is shown that road construction affects mangrove ecosystems and has an influence on microbial diversity. The decrease in bacterial diversity in the degraded mangrove sediments was established. The authors demonstrated modern methods of research using the whole metagenomic sequencing approach with more samples. The article will be interesting to read for specialists and may be accepted for publication in present form.
Author Response
Dear reviewer,
Thank you for accepting to review our manuscript. The second reviewer suggested some minor modifications, which we made and are described in the final version.